# Diversity of Lacewings (Neuroptera) in an Altitudinal Gradient of the Tacaná Volcano, Southern Mexico

**DOI:** 10.3390/insects13070652

**Published:** 2022-07-19

**Authors:** Rodolfo J. Cancino-López, Claudia E. Moreno, Atilano Contreras-Ramos

**Affiliations:** 1Posgrado en Ciencias Biológicas, Unidad de Posgrado, Circuito de Posgrados, Universidad Nacional Autónoma de México, Mexico City 04510, Mexico; cancinorodolfoj@gmail.com; 2Colección Nacional de Insectos, Departamento de Zoología, Instituto de Biología, Universidad Nacional Autónoma de México, Mexico City 04510, Mexico; 3Centro de Investigaciones Biológicas, Universidad Autónoma del Estado de Hidalgo, Mineral de la Reforma, Hidalgo 42184, Mexico; cmoreno@uaeh.edu.mx

**Keywords:** taxonomic diversity, taxonomic distinctness, species composition, beta diversity, species turnover, nestedness, elevational gradient

## Abstract

**Simple Summary:**

Lacewings are insects with a great diversity of morphology and habits and are potentially important as bioindicators and biological control agents. However, there is little worldwide information on their patterns of distribution and diversity. Therefore, it is essential to understand what changes occur in their diversity through spatial changes such as elevation. We analyzed Neuroptera diversity locally and between sites through an elevation gradient, finding two marked trends: (1) a decrease in richness with increasing altitude and, (2) greater diversity and potential richness at an intermediate elevation. In addition, a high degree of species turnover means that there is an evident transition between the lowland communities and the forest in the upper parts of the volcano, reflecting an altitudinal replacement of species and exclusivity for certain altitudes. These patterns will help us understand the distribution diversity of lacewings for better management and conservation of insects and the ecosystems with which they are associated.

**Abstract:**

Neuroptera is an order of insects with a moderate diversity of species numbers yet a high between-family morphological diversity, which has a significant ecological role as a predator. However, there are few studies focused on describing changes in species diversity along environmental gradients. We evaluated changes in the alpha and beta diversity of species and the higher taxa in Neuroptera communities in the Tacaná Volcano in southern Mexico. Five sites each at different altitudes were studied through systematic annual sampling. The taxonomic and phylogenetic alpha diversity were analyzed, as well as the beta diversity and its components, species turnover and nestedness. The alpha diversity had two trends: (1) decreased standardized richness and taxonomic distinctness with increasing altitude, and (2) increased estimated richness and species diversity at intermediate altitudes. The highest turnover values for species, as well as for supra-specific taxa, were recorded at sites with lower altitudes. The highest total beta diversity value was recorded at elevations above 3000 m, whereas the highest number of species and supra-specific taxa were observed at sites between 600 and 2000 m, with an evident decrease above 3000 m. The type of vegetation and environmental conditions may be influencing the decrease in diversity toward higher elevations, which could explain the niche specialization of Neuroptera species to particular sites within the gradient. These results highlight the need to study the environmental factors and their effects on species composition along an elevation gradient.

## 1. Introduction

Biodiversity patterns across spatial gradients have long been a topic of great interest for understanding evolutionary processes that shape biological communities [1,2]. In particular, altitude gradients allow us to analyze changes in species richness and community composition in response to environmental variations such as precipitation, temperature, vegetation structure, and humidity, among others [3,4]. For this reason, mountain systems have been considered natural laboratories for the analysis of different patterns, which can explain the distinct ecological processes that shape the attributes of biodiversity, since environmental conditions change with an increasing elevation over short spatial distances, which influences the distribution of species [5,6].

In general, species richness along elevational gradients follows two main patterns: decreasing richness at higher altitudes, or greater richness at intermediate elevations [7,8]. In the first case, extreme climates at high altitudes harbor few species capable of tolerating such conditions, and at low altitudes, a greater number of species are concentrated due to the stability of climatic conditions [9]. On the other hand, the pattern of higher richness at intermediate altitudes is related to the mid-domain effect, which consists of an increasing overlap of species toward the center of a spatial domain due to the presence of strong spatial boundaries at the upper and lower edges, regardless of the influence of the relationship between species and the environment [1,10].

Beta diversity (dissimilarity in species composition) increases with increasing altitude, i.e., there is higher dissimilarity at mountain tops [5,6,8,11], with some exceptions of an opposite pattern [12,13]. When total dissimilarity is partitioned into dissimilarity due to turnover (species replacement between sites) and dissimilarity due to nestedness (species loss or gain between sites) [14], species turnover is often the main component that explains total dissimilarity [5,8,15]. However, there are cases in which high nestedness values have been recorded at high elevations [2].

Neuropteran communities can be excellent models for investigating the effects of environmental and spatial changes on species diversity and habitat composition [16] because differences in habitat types and shapes can determine and influence the diversity, abundance, and distribution of Neuroptera in different ecosystems [17]. Neuroptera is a group of mainly terrestrial insects with approximately 6000 species, 603 genera, and 15 families [18], with a worldwide distribution. Neuroptera species occupy a wide variety of habitats, from tropical to temperate. They present a morphological diversity and variety of specializations in their life histories, mainly during their larval stage [19]. Some adults may feed on plant structures (such as honeydew and nectar from flowers), but most, and primarily the larvae, are predators of many soft-bodied arthropods including aphids, whiteflies, small lepidopteran caterpillars, and eggs, or as parasitoids of some spiders, bees, and wasps, with some families associated with termites or river sponges [17]. Neuropterans may serve as indicator groups for habitat transformation [20] as they present a high specificity to particular habitats and biomes, making them sensitive to ecological alterations [21,22]. Many species depend on external thermal conditions, such as wind speed and ambient temperature, to maintain viable populations, using these variables as predictors of species richness and abundance [17,23].

In many cases, the composition and characteristics of neuropteran communities are determined by the species of prey and their abundance, the microclimates present, and the plant structure of the habitat [17,24]. In some families, such as Chrysopidae and Hemerobiidae, their high capacity to colonize and adapt to new environmental conditions is evident [25], which allows them to adjust their feeding areas based on their habits, cost-benefits, and abilities to locate their food resource [26]. Despite this, Neuroptera are generally insects with a weak ability to fly (except for Ascalaphinae), which means that their dispersal ability often depends on air currents [27]. In general, Neuroptera species seem to prefer the presence of shelters and food resources that allow them to inhabit different environments, with the need for a complex plant physiognomy that provides them with diversified niches for their survival [28].

Studies on the alpha diversity and community structure of Neuroptera have been carried out using different approaches, and their study has increased in recent years. Some aim to understand the differences in diversity between sites, the local diversity of a site, or the diversity of communities over time (temporal). Regarding beta diversity, studies have focused mainly on the similarity between areas, types of ecosystems, and families, reporting significant differences between communities at different sites [26,29]. For Neuroptera, beta diversity components have been little studied, although it has been noted that in communities of lacewings, there might be a high species turnover.

One of the factors that few studies have explored is the effects of altitude on the diversity of Neuroptera, whereas some studies have focused on the composition and diversity of the different families. It has been observed that certain groups seem to be restricted to particular areas, from zones with cold temperatures (alpine areas) to zones with warmer temperatures [30]. In recent works, the different factors associated with the abundance and diversity of neuropterans have been analyzed. It has been reported in families such as Myrmeleontidae (without Ascalaphinae) and Nemopteridae, that the abundance increased with altitude, whereas other families such as Chrysopidae, Coniopterygidae, and Ascalaphinae decreased with elevation [29]. Despite this, elevation does not seem to influence families in the same way and, in many cases, trends change depending on the geographical location. Also, a recent study showed that the alpha diversity of Chrysopidae decreased with increasing elevation, as well as the component that best explains the dissimilarity changes along the altitudinal gradient. They reported that nestedness replaces species turnover with increasing altitude [31]. These patterns depend on the biological/ecological requirements of Neuroptera species and factors such as temperature and seasonality [32]. It is crucial to continue carrying out studies on how the communities of neuropterans change with altitude, as different environmental factors change with elevation and may function as environmental filters that affect the diversity or the species distribution.

In the present study, we analyze the diversity of Neuroptera (lacewings and allies) along an altitudinal gradient of the Tacaná Volcano Biosphere Reserve, which lies at the northernmost limit of the mountainous area called the Central American Nucleus, which is part of the Mesoamerican Biological Corridor in southern Mexico. This region presents a high biological richness, endemism, and a great variety of vegetation types, resulting from the assemblage of biotas of Nearctic and Neotropical origin [33]. This study aimed to analyze the changes in the Neuroptera communities along an altitudinal gradient of the Tacaná Volcano. Therefore, the following specific objectives were proposed: (1) estimate the potential number of species at the local and regional levels (Tacaná Volcano) to assess the completeness of the inventory; (2) analyze the alpha diversity of species and taxa (taxonomic distinctness) along the altitudinal gradient; and (3) evaluate the beta diversity (dissimilarity) and its turnover and nestedness components due to differences in species richness, both for species and for higher taxa through the altitudinal gradient. Higher values of species richness and diversity were expected to be found at mid-elevations according to the mid-domain effect, due in part to spatial limitations at high altitudes (reduction in area) and low altitudes (reduction of conserved zones because of anthropogenic activities). Regarding beta diversity, a higher dissimilarity in species composition is expected between sites with high and low altitudes compared to the intermediate elevation sites; areas at medium elevation present similar environmental conditions (same type of vegetation), which may lead to the recording of similar faunas. However, the phylogenetic alpha and beta diversities could decrease with altitude, with a higher number of lineages at low altitudes better adapted to stable environmental conditions, and fewer lineages adapted to the extreme conditions at the upper parts of the volcano, which suggests an environmental filter that may influence the dispersal or colonization of lineages along the gradient.

## 2. Materials and Methods

### 2.1. Study Area

The Tacaná Volcano Biosphere Reserve is located in the state of Chiapas, Mexico, and in the department of San Marcos, Guatemala. The volcano reaches an altitude of 4092 m and has an area of approximately 300 square kilometers, of which three-quarters correspond to Mexico, and contains a wide variety of types of vegetation (predominantly cloud forest). The volcano is part of the Central American Volcanoes and Chiapas Coastal Plain Subprovince [34], a unit extended along the Pacific between the Isthmus of Tehuantepec and Guatemala. In addition, it belongs to the Mexican Transition Zone within the Altiplano de Chiapas biogeographic province [35]. The environmental heterogeneity of the volcano offers a wide spectrum of habitats and conditions, making it an ideal study area to understand patterns of diversity across an elevational gradient in the Neotropics.

The reserve presents an average annual rainfall of 4438 mm with a relative humidity of 90% during the rainy season, autumn, and part of the winter, whereas during the dry season, it remains above 50%. Taking into account the Köppen climate classification modified by García [36], the climates that predominate in the reserve are the following: temperate humid (average annual temperature of 15.3 °C), semi-warm humid (average annual temperature of 20.7 °C), and warm humid (average annual temperature of 24.3 °C) with abundant rains in summer.

### 2.2. Sampling Design and Method

Five sampling sites were located each at a different altitude within the part of the volcano belonging to Mexico from 661 to 3246 m above sea level (Figure 1) in the municipalities of Cacahoatán and Unión Juárez in Chiapas state (Appendix A). Systematic monthly samplings were carried out for a year (February 2018 to January 2019) at the five sampling sites. The collection period in each site was two and a half days each month during days with less moonlight. However, Malaise traps were working permanently throughout the year (with samples picked up monthly). Each month, seven sampling points separated by 500 m were established at the sampling sites where different sampling techniques were placed: a black light trap (bucket) and a black and white light trap (screen) (sampling point 1), two Malaise traps (sampling point 2), one ground-level intercept trap, and one yellow plate trap hung on the tree canopy (both placed at each of the remaining five sampling points) [37,38,39]. All points were randomly placed in a sampling area of approximately 2 km^2^. In addition, sweeping was applied to the surroundings of the seven sampling points on the canopy and the herbaceous stratum of the vegetation of each sampling site with the use of an entomological net for four hours (10:00–14:00) per person (2 people) [39] (Figure 2). All the specimens collected through the different techniques during the twelve months of fieldwork were considered as a single annual sample unit for each site; in this regard, the temporal variation of diversity was not analyzed in this work. The different traps were placed at a minimum distance of 200 m between them. Specimens were identified and deposited in the Colección Nacional de Insectos of the Instituto de Biología of the Universidad Nacional Autónoma de México (CNIN-UNAM), Mexico. It is necessary to mention that this article stems from a global project on the Tacaná Volcano Neuroptera. The data used in this investigation were exclusively those obtained from the collection methods with annual systematic sampling carried out by Cancino et al. [39], without taking into account material from museum collections or other sites within the volcano, as mentioned in the previous publication.

### 2.3. Data Analysis

#### 2.3.1. Inventory Completeness Estimation

The estimations of the completeness of the inventory and the potential number of species for each altitude level were calculated using the sample coverage (Sc) estimator [40]. These estimates were carried out using the iNEXT version 1.3 program [41]. These data were randomized 100 times and compared with the observations [40] with a confidence interval of 95%.

#### 2.3.2. Alpha Diversity: Species and Taxa

Hill’s numbers were used for the analysis of species diversity, either of order 0 (richness of species), order 1 (diversity of rare and common species), or order 2 (diversity of dominant species) according to Jost [42]. These estimates were carried out using the iNEXT version 1.3 program [41]. The analyses were carried out with 100 randomizations and were extrapolated to twice the number of samples [40] with a confidence interval of 95%. To compare the different diversity values between sites, the results were standardized to the same sample coverage (Sc), which indicates the proportion of the total community represented by the sampled species [40] using the iNEXT program. The calculated diversities were compared using 95% confidence intervals [43]. A visual comparison was made based on the superposition of the upper and lower intervals in order to establish whether significant differences between the values of each of the sites exist [44].

For the analysis of alpha phylogenetic diversity, the proposal of Clarke and Warwick [45] was used, which is based on the average taxonomic distances (length of the taxonomic routes) between two randomly selected species in the Linnaean hierarchy, which includes all species in an assemblage. For this, an abundance matrix was used as well as a second matrix with the hierarchical classification of all the species. Three indices of phylogenetic alpha diversity were obtained: (1) taxonomic distinctness (DivT: Δ*), which expresses the total taxonomic distance between two randomly chosen species (with replacement), (2) average taxonomic distinctness (DisT: Δ+), which represents the average of taxonomic distances between species, and (3) the taxonomic variation (VarT: Λ+), which measures the variance of the taxonomic distances between species [45]. These indices were compared with a null model built from 1000 randomizations of the set of species of each community in the PRIMER v7 Trial version [46], in order to assess whether the values obtained are statistically different from those expected at random.

#### 2.3.3. Beta Diversity: Species and Taxa

The total taxonomic beta diversity on the volcano was evaluated with the Sorensen index (βSOR) and was analyzed with its two components: the dissimilarity due to turnover (βSIM) and the dissimilarity due to differences in richness (nestedness) (βNES) under the multiple-site approach [14]. In addition, under the pair-wise approach, the beta diversity between consecutive sites was measured through the elevation gradient (βsor = βsim + βnes), calculating the relative contribution of each component (in percentages) based on the incidence of the species [2,8].

Phylogenetic beta diversity (total dissimilarity of taxa) was analyzed using the incidence (presence–absence) of the taxa present at the different sites. Analogous to the beta diversity of species, the total dissimilarity between the taxonomic structures of the communities was measured using the Sorensen index. The approximation of Bacaro et al. [47] and Baselga [14] is a taxonomic dissimilarity method from the taxonomic distinctness approach of Clarke and Warwick [45]. This approach compares species richness and variations in taxonomic structures between assemblages, where all taxa have the same level of importance, regardless of their hierarchical levels. The beta diversity of taxa and its taxa turnover and nestedness components was calculated using the multiple-site approach and consecutive pairs of sites [14]. All beta diversity analyses were performed with the Betapart v.13 package in the R program [48].

## 3. Results

A total of 2527 individuals corresponding to 105 species, 28 genera, seven tribes (Chrysopini, Leucochrysini, Coniopterygini, Conwentzini, Fontenelleini, Myrmeleontini, and Ululodini), 13 subfamilies (Chrysopinae, Coniopteryginae, Aleuropteryginae, Hemerobiinae, Megalominae, Microminae, Notiobiellinae, Sympherobiinae, Mantispinae, Calomantispinae, Myrmeleontinae, Ascalaphinae, and Symphrasinae), and six families of Neuroptera were collected. The highest number of species was represented by Chrysopidae (51 species) and Hemerobiidae (29 species), and the greatest abundance by Hemerobiidae (709 individuals) and Coniopterygidae (1094 individuals) (Table 1). Forty-four species occurred at only one site, and only two species were found at all five sampling sites; On average, Neuroptera species were recorded at two sampling sites (Table 1). At each site, most species were in low abundance and very few species were dominant (Figure 3).

### 3.1. Inventory Completeness

The mid-elevation sites (between 1200 and 2000 m) recorded more than 70% of the estimated species, whereas the lower and higher elevations recorded less than 70% of the estimated species (Table 1). At the regional level, 140 species were estimated for the Tacaná Volcano, so the 105 recorded species represent 73% completeness of the inventory. Therefore, based on the sample coverage estimator, 35 species of Neuroptera could still be recorded on the volcano with the same collection techniques. A low number of rare species was found at the regional level; 24 species had only one individual (22.8% of the total) and 11 species had two individuals (10.5% of the total).

### 3.2. Alpha Diversity: Species and Taxa

The sample coverage of each site was high, with values above 90%. When standardizing species richness to the same estimated sample coverage (Sc = 0.970), only the last elevation at the highest altitude had lower richness and diversity than the lower sites according to the confidence intervals. Standardized richness decreased with increasing altitude (Figure 4A). On the other hand, the observed and estimated richnesses and order 1 and 2 diversities had higher values at intermediate altitudes (Figure 4B,C). The taxonomic distinctness (DivT) in the first four sites presented similar values. The highest value is observed at site three and the lowest value at site five. The average taxonomic distinctness (DisT) values showed a similar trend, with the highest value at site three and the lowest value at site five. Finally, the taxonomic variation (VarT) is lower in sites four and five, whereas the highest value was found for site two (Figure 4D). Therefore, the taxonomic structure remains relatively constant at low and medium altitudes, with a drastic decrease at altitude ranges above 3000 m a.s.l.

### 3.3. Beta Diversity: Species and Taxa

The taxonomic beta diversity (total dissimilarity) among the five sampling sites is high (βSOR = 71.4%) and is mainly due to species turnover (βSIM = 63.3%) with a low nestedness contribution (βNES = 8.1%). When comparing the species composition between consecutive sites in altitude along the gradient, it was found that among the first four sites of the altitude gradient (between 600 and 2000 m), the beta diversity is mainly due to the turnover of species, and the highest turnover occurs between 600 and 1000 m a.s.l. (Table 2). However, the highest total beta diversity occurred between sites four and five (from 2000 to 3000 m), mainly due to nestedness (Figure 5A). Thus, sample site five has a neuropteran fauna that is mostly a subset of the fauna found at site four; of the 16 species recorded at site five, only 4 were found exclusively at this elevation (*Ungla* sp.1, *Megalomus* sp.1, *Sympherobius* sp.1, and *Hemerobius alpestris* Banks, 1908), whereas the remaining 12 species were also found at site four.

Phylogenetic beta diversity, which took into account the supra-specific taxonomic levels between the five sites of the altitudinal gradient (βSOR = 60.4%), was explained by a high turnover (βSIM = 49.8%) and a low nestedness (βNES = 10.6%). When evaluating the taxonomic dissimilarity between pairs of sites, a very similar trend was found for the beta diversity of species, although with lower values for including supra-specific taxa (Figure 5B).

## 4. Discussion

Neuroptera communities have been reported with low abundances compared to other insect orders [38,49]; however, a high number of individuals and species were reported in this study. The presence and representativeness of families such as Chrysopidae, Hemerobiidae, and Coniopterygidae is not a different trend from that previously recorded in other studies where they appear as the most abundant or richest families of Neuroptera [26,29,49,50].

Abundance presented the highest values at medium and high altitudes, possibly related to the restricted distribution of some families along the gradient. As Bozdogan [51] points out, the abundance of certain families increases with altitude and decreases for others. For example, Hemerobiidae has higher abundance values at high altitudes (with evident adaptations to extreme environmental conditions), whereas Chrysopidae has higher abundance values at low and medium elevations (with a preference for more tropical or warmer areas) [30,32]. In this work, a decrease in the abundance of Chrysopidae, Mantispidae, Myrmeleontidae, and Rhachiberothidae was observed with increasing altitude, in contrast to families such as Hemerobiidae and Coniopterygidae. Therefore, we believe that the abundance of species may be influenced by factors such as their biology, vegetation, climatic conditions, and anthropogenic activity. This agrees with what was observed in a study on the elevation diversity patterns of Chrysopidae, suggesting that temperature had a significant effect on the abundance of these green lacewings [31], reducing their abundance in areas with lower temperatures.

A particular dominant species was reported per site (*Semidalis soleri* (site 1), *Coniopteryx simplicior* (site 4), *Hemerobius discretus* (Site 5)), in some cases shared between nearby sites (*Semidalis problematica* (sites 2 and 3) (Figure 3). This has been observed in lacewing communities where the species are correlated to different habitats (mainly the dominant ones), characterizing the habitats by the occupancy of dominant species or exclusivity [21]. Also, Coniopterygidae seems to select their habitat depending on specific plant substrates, which are sometimes local and rare and sometimes extremely abundant [52]. Based on previous studies, the specificity of a plant substrate could explain the presence of dominant species at specific sites so that sites at different levels that shared dominant species also had the same type of vegetation. On the volcano, the dominant species belonged to the families Hemerobiidae and Coniopterygidae, which have been previously recorded in other studies as abundant [49,50].

### 4.1. Inventory Completeness

The lack of species to be recorded to complete the faunal inventory of the volcano is possibly influenced by the extension of the volcano (by only recording species present in the Mexican section), the high degree of specificity at certain altitudes, and the need for intense efforts for sampling. Several authors discuss the difficulties of sampling and that some require greater efforts both in the time needed and the methods used [49,53]. Due to the fact that the populations present relatively low abundances, the choice of methods also has a clear influence on the characteristics of the samples obtained [37,38].

### 4.2. Alpha Diversity: Species and Taxa

Studies on the diversity of the Neuroptera have focused on agroecosystems and differences between different types of habitats [50,54] but few have evaluated changes in species composition along an elevational gradient [17,31,51].

The values of the estimated diversities did not show significant differences, except for sites four and five (sites above 2000 m). The values of diversity q0 (species richness) and q1 (diversity of common and rare species) decreased with increasing altitude (Figure 4A,B). This was also observed in a study by Lai et al. [31] where the alpha diversity of Chrysopidae decreased with increasing elevation. In the cases of diversity q2 (diversity of dominant species), the number of dominant species was similar at the first four sites in contrast to site five, which had the lowest value (Figure 4C). Both the composition and characteristics of Neuropteran communities are often determined by prey species and abundance, microclimate, and plant structure [17,26,51]. These changes in the values of the diversities in the study sites could be due to the heterogeneity in the plant structures, where the lowest altitude site presented a disturbed plant physiognomy and patches of agroecosystems, whereas medium altitudes were characterized by the presence of cloud forests and coffee plantations. On the other hand, the highest altitudes above 3000 m, had extreme environmental conditions with a loss of vegetation cover and the presence of pine forests and oak patches.

This means that at high altitudes, the richness and diversity of Neuroptera are exclusive to those species that adapted to the extreme conditions. Therefore, the increase in elevation can affect the distribution of Neuroptera species [55]. Therefore, the dispersal capacity of the species and their local abiotic conditions, such as temperature, precipitation, and wind speed, among others, can behave as filters, which generates differences in the composition of species in different areas [56]. This pattern of higher richness at low and medium altitudes has been observed in other groups of insects such as aquatic invertebrates, ants, wasps, and bees, among others [5,8,57].

The values of the alpha diversity based on the degree of species relatedness were the highest at the sites between 600 and 1700 m (Figure 4D). This suggests that these areas had greater diversity in their taxonomic structure as a reflection of greater phylogenetic separation between the species that make up these communities. The high values of taxonomic variation show that most species are concentrated in a few supra-specific taxa [45,58]. Sites with ranges above 2000 m had lower values of taxonomic distinctness, showing low diversity and taxonomic difference, which indicated that the species of Neuroptera are better distributed in the different hierarchical levels present in these communities. Although there is no decrease in the diversity of higher taxa with increasing altitude, there is a decrease in diversity at high altitudes, as has been observed in other studies where phylogenetic diversity decreases with increasing elevation [59,60,61].

Both the number and abundance of each species and the variety of taxa present in the community seem to decrease at high altitudes, although the distribution of species is better represented than at sites at low or medium altitudes. This diversity of taxa in Neuroptera is represented at the family level where families such as Myrmeleontidae and Mantispidae diversify better at low or medium altitudes but their presence and diversification decrease as the altitude increases. On the other hand, families such as Hemerobiidae seem to increase their numbers and diversify with increasing elevation.

Finally, the changes in the variety of taxa between communities could be influenced by the adaptations and life histories of the different lineages that comprise them. As an example of this, the family Chrysopidae has greater diversity at low and medium altitudes but has little representation at altitudes above 3000 m; although together with Hemerobiidae, they are known for their great capacity for colonization and adaptation to new conditions [62]. In the case of Chrysopidae, some genera are frequently reported in agroecosystems, which, together with the native vegetation, provide high availability of food, niches to occupy, and adequate climatic conditions. On the other hand, Hemerobiidae species seem to diversify better at high elevations because their adaptations and life histories allow them to colonize habitats with more extreme conditions [63] and possibly avoid competition with Chrysopidae lineages. This showed possible distinct tendencies for the different families, although at a global level, the diversity tendencies are the same at the species and higher taxa levels.

### 4.3. Beta Diversity: Species and Taxa

At the regional level, the beta diversity (the change in species diversity from one site to another) showed a strong turnover pattern as the component that had the greatest contribution along the gradient. For Neuroptera, there is one study that uses this approach to investigate its influence on changes in species composition. On the other hand, in other groups of arthropods, these components have been evaluated where the turnover of species is presented as the most important component [2,8].

When we compared diversity between sites, almost all comparisons were explained by species turnover (Table 2), except for the two sites above 2000 m, which can be explained mainly by nestedness (Figure 5). Previously, this has been recorded in Chrysopidae in an altitudinal study, where nestedness replaced turnover as the main component of dissimilarity as the elevation increased (mainly in sites with low temperatures) [31]. Also, nestedness values at high altitudes have been reported in beetle communities in a mountainous system of Colombia, where high turnover values were reported at a general level but at sites above 2000 m, the beta diversity was better explained by nestedness [2]. This is probably because the conditions that exist at these altitudes function as environmental filters that do not allow the colonization of other species. In addition, in other studies, a latitudinal pattern has been recorded where above the 37th parallel, the beta diversity is due to a nestedness pattern, and to the south, the turnover is more important [64], which suggests that this tendency could be repeated for certain groups of insects in an altitudinal gradient.

The fact that the changes in the composition of Neuroptera in this study were mainly due to turnover leads us to hypothesize that this turnover is due to the selection of species in a certain environment or due to dispersal processes [65]. In addition, it is known that the structure, vegetation cover, and climatic conditions (such as temperature and wind speed) are important factors for the presence of certain species [17,21,23,66]. The first sites in this study (high turnover values) also have large extensions of conserved forests and patches of agroecosystems that could be functioning as a means of safeguarding biodiversity. This could allow them to have more specific niches for their diversification and generate the stratification of species along the altitudinal gradient.

The dissimilarity of Neuroptera on the volcano between the different sites was high as previously reported, with high dissimilarities between sites within a wide distance or with different environmental characteristics [30,51]. Moreover, in other cases where the sites presented the same environmental or vegetation conditions and were close to each other, they had a low dissimilarity [50,67].

The difference in taxa compositions between communities was better explained by the turnover of supra-specific taxa since particular genera were substituted at the sites. Furthermore, nestedness seems to have a strong effect at sites above 2000 m. Both total and turnover dissimilarity values were low compared to species-level values due to the low supra-generic diversity compared to the high number of species present on the volcano. Therefore, it seems that the differences in the taxonomic structures between communities in the altitudinal gradient are more diversified at altitudes below 1800 m, whereas above this range, the diversification begins to decrease.

Finally, the variety of taxa of the Neuroptera community along the Tacaná Volcano shows a strong pattern of species turnover. This means that there is a strong transition between the lowland communities and the forest in the upper area of the volcano that generates an evident altitudinal replacement of species and a clear exclusivity for certain altitudes.

## 5. Conclusions

Neuroptera species presented an evident restriction to particular sites, with few families distributed throughout the altitudinal gradient (Chrysopidae, Coniopterygidae, and Hemerobiidae). Generally, these insects presented low abundance on the volcano. The highest abundance peaks were observed at medium and high altitudes. The particular dominant species for each site are possibly associated with the environmental conditions and vegetation types. The need to increase the sampling effort at the local level was also observed, mainly focused on groups less represented in the study and with specific requirements at the time of collection. The highest estimated species richness value was recorded at low altitudes, decreasing with increasing elevation. The values of diversities q1 and q2 have similar trends, showing a decrease with increasing altitude, with the highest value at the site above 1000 m; the lower diversity value at low altitudes (>1000 m) is possibly due to the anthropogenic effect. The alpha diversity based on the degree of species relatedness showed that the diversity in the taxonomic structure seems to remain constant at low and medium altitudes, with a drastic decrease in altitude ranges above 3000 m. The high altitudes had better species distribution in the different hierarchical levels. Total species dissimilarity values at the local and regional levels show strong species turnover along the altitudinal gradient, except for sites above 2000 m, which were better explained by nestedness. The most evident turnover was between high and low altitudes. The difference in the taxa composition between communities recorded a global value of 71%. On the other hand, the beta taxonomic distinctness recorded a similar trend to that calculated for species but with much lower turnover values for the supra-specific taxa both regionally and between sites. These results support the influence of changes in elevation on the diversity and composition of Neuroptera species, which may be influenced by mechanisms such as environmental factors or species dispersal limitations (reflected by high turnover rates).

## Figures and Tables

**Figure 1 insects-13-00652-f001:**
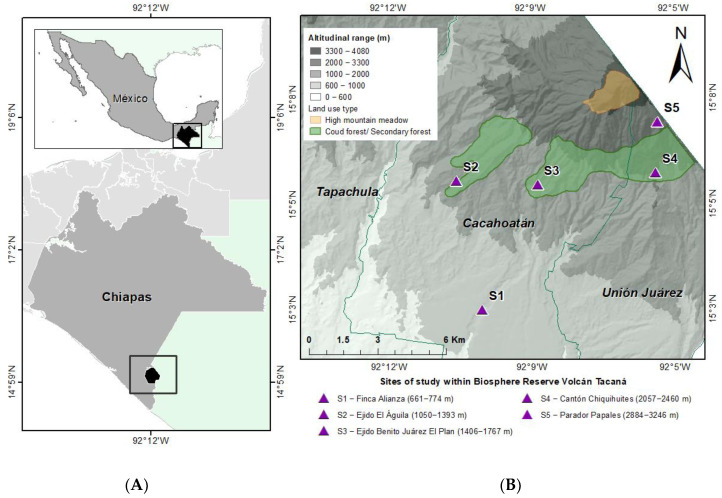
Map of the distribution of sampling sites in the Tacaná Volcano Reserve, Chiapas, Mexico. (**A**) Geographical Location, (**B**) Sites of study.

**Figure 2 insects-13-00652-f002:**
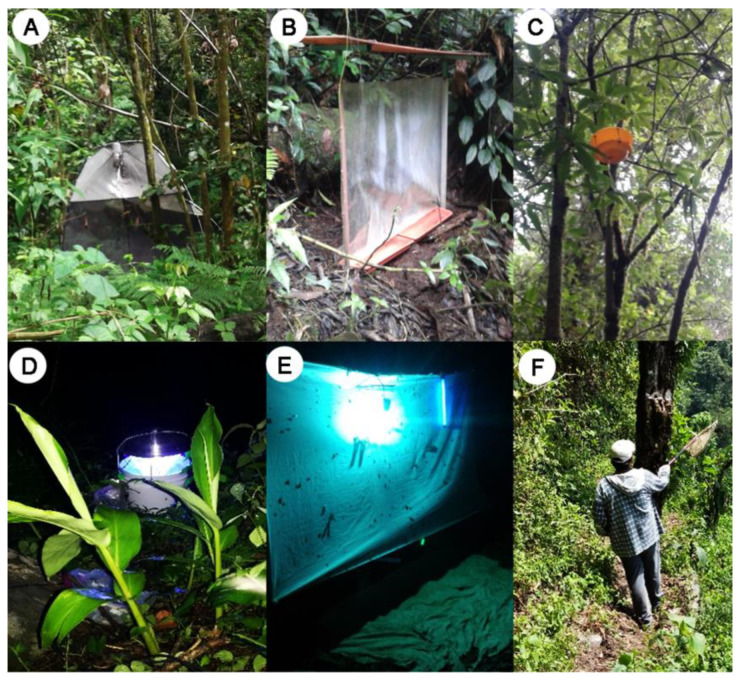
Different types of sampling methods implemented in this study. (**A**) Malaise trap; (**B**) Ground-level interception traps; (**C**) Yellow plate traps; (**D**) Black light trap; (**E**) Black and white light trap; (**F**) Entomological net (Reprinted with permission from Cancino et al. [39]. Copyright 2021, by the authors).

**Figure 3 insects-13-00652-f003:**
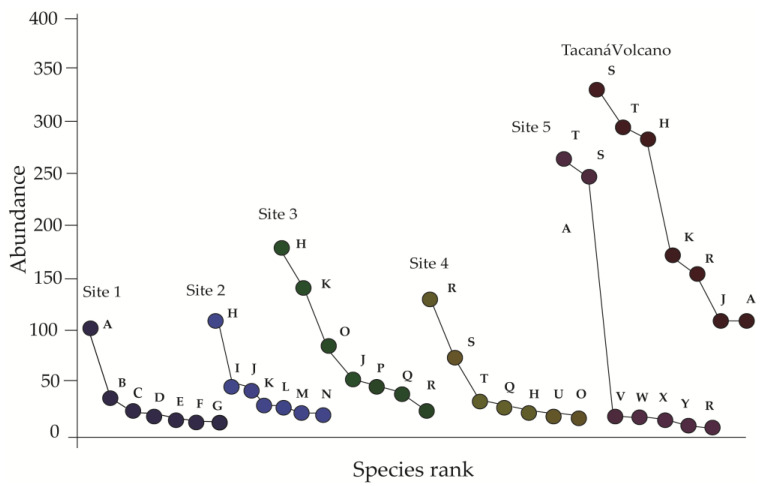
Rank-abundance curves of Neuroptera species for the different sampling sites and at the regional level of the Tacaná Volcano, Mexico. Species: A. *Semidalis soleri*, B. *Leucochrysa pretiosa*, C. *Leucochrysa askanes*, D. *Megalomus minor*, E. *Leucochrysa lateralis*, F. *Myrmeleon timidus*, G. *Leucochrysa tarini*, H. *Semidalis problematica*, I. *Chrysopodes crassinervis*, J. *Ceraeochrysa sarta*, K. *Hemerobius hernandezi*, L. *Semidalis hidalgoana*, M. *Leucochrysa maculosa*, N. *Ceraeochrysa arioles*, O. *Meleoma titschacki*, P. *Chrysopodes varicosus*, Q. *Ceraeochrysa tacanensis*, R. *Coniopteryx simplicior*, S. *Hemerobius jucundus*, T. *Hemerobius discretus*, U. *Semidalis manausensis*, V. *Coniopteryx latipalpis*, W. *Conwentzia barretti*, X. *Hemerobius alpestris*, Y. *Sympherobius axillaris*.

**Figure 4 insects-13-00652-f004:**
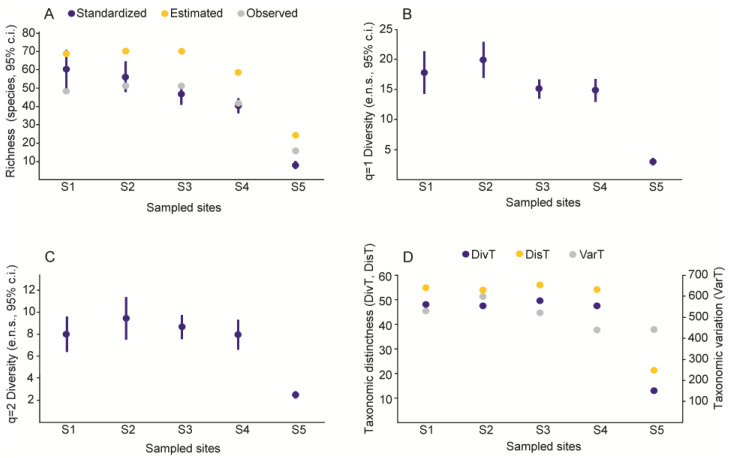
Species richness and diversity of orders 1 and 2 and taxonomic distinctness indices for each of the sampled sites. (**A**) Species richness, (**B**) Order q = 1 diversity, (**C**) Order q = 2 diversity, (**D**) Taxonomic distinctness (DivT), average taxonomic distinctness (DistT), and taxonomic variation (VarT). For standardized diversity values, error bars are 95% confidence intervals (c.i.), and for q = 1 and q = 2 diversities, the units are the effective number of species (e.n.s.).

**Figure 5 insects-13-00652-f005:**
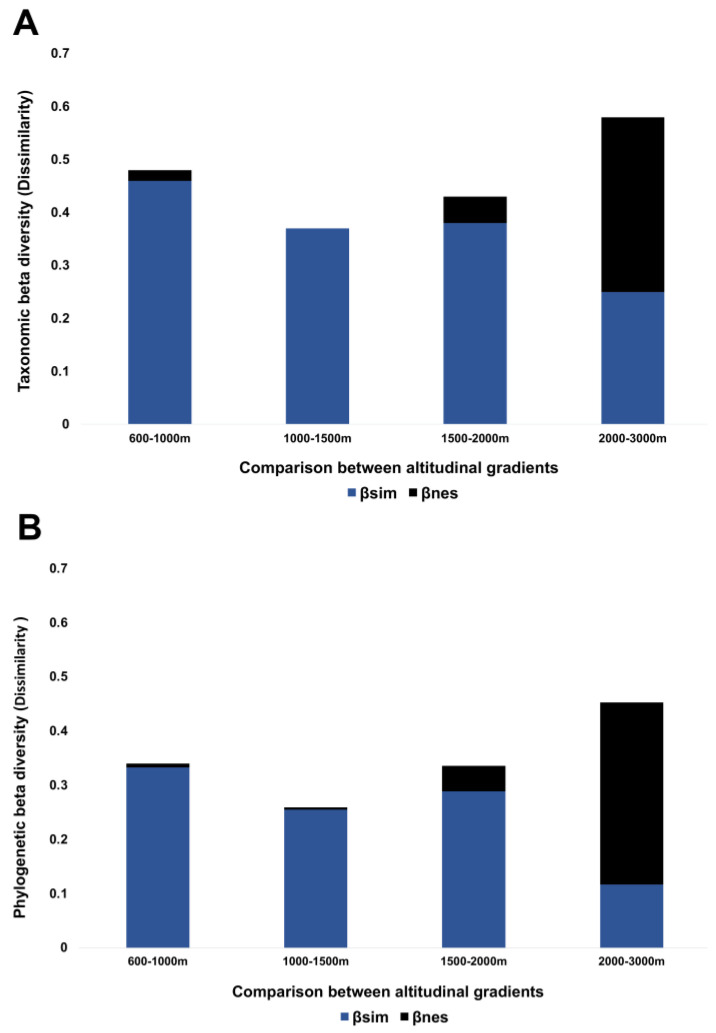
Taxonomic and phylogenetic beta diversity (Dissimilarity [βsor]) of the Neuroptera community of the Tacaná Volcano with the criterion of pairs of sites, with each of its components, turnover (βsim), and nestedness (βnes). (**A**) Taxonomic beta diversity across the altitude gradient of the Tacaná Volcano obtained using species incidence. (**B**) Phylogenetic beta diversity through the altitudinal gradient of the Tacaná Volcano using taxa incidence.

**Table 1 insects-13-00652-t001:** List of species of Neuroptera, number of sampling sites where the species was found, and abundance at each sample site along the altitudinal gradient at the Tacaná Volcano, Mexico. S1–S5: Site 1 to Site 5, where S1 has the lowest and S5 the highest altitude. * Species previously identified as *Ululodes* sp.1 (Adapted with permission from Cancino et al. [39]. Copyright 2021, by the authors).

	Scientific Name							
Family		Number of Sites Occupied	S1	S2	S3	S4	S5	TOTAL
	Genus *Ceraeochrysa*		661–774 m	1050–1393 m	1406–1767 m	2057–2460 m	2884–3246 m	
Chrysopidae	*C. achillea* de Freitas & Penny, 2009	2	7	1	0	0	0	8
	*C. arioles* (Banks, 1944)	4	1	11	1	2	0	15
	*C. cincta* (Schneider, 1851)	3	1	3	0	1	0	5
	*C. cubana* (Hagen, 1861)	2	2	0	0	1	0	3
	*C. defreitasi* Penny, 2002	1	0	0	1	0	0	1
	*C. derospogon* de Freitas & Penny, 2009	2	0	2	1	0	0	3
	*C. effusa* (Navás, 1911)	3	2	3	14	0	0	19
	*C. infausta* (Banks, 1945)	2	2	0	0	1	0	3
	*C. lineaticornis* (Fitch, 1855)	2	0	6	1	0	0	7
	*C. sanchezi* (Navás, 1924)	2	1	2	0	0	0	3
	*C. sarta* (Banks, 1914)	4	5	37	48	11	0	101
	*C. squama* de Freitas & Penny, 2001	2	1	1	0	0	0	2
	*C. tacanensis* Cancino & Contreras, 2019	3	0	2	35	26	0	63
	*Ceraeochrysa* sp. 1	2	0	1	1	0	0	2
	Genus *Chrysoperla*							
	*C. asoralis* (Banks, 1915)	2	0	2	0	1	0	3
	*C. externa* (Hagen, 1861)	1	1	0	0	0	0	1
	Genus *Chrysopodes* Subgenus *Chrysopodes*							
	*C. crassinervis* Penny, 1998	2	0	41	3	0	0	44
	*C. varicosus* (Navás, 1914)	4	1	8	42	2	0	53
	*Chrysopodes* sp. 1	1	0	1	0	0	0	1
	*Chrysopodes* sp. 2	1	0	0	1	0	0	1
	Genus *Leucochrysa* Subgenus *Leucochrysa*							
	*L. clara* (McLachlan, 1867)	2	0	6	2	0	0	8
	*L. colombia* (Banks, 1910)	2	0	0	1	1	0	2
	*L. pretiosa* (Banks, 1910)	1	36	0	0	0	0	36
	*L. varia* (Schneider, 1851)	1	0	1	0	0	0	1
	*L. variata* (Navás, 1913	3	1	2	1	0	0	4
	Subgenus *Nodita*							
	*L. amistadensis* Penny, 2001	2	0	0	1	2	0	3
	*L. askanes* (Banks, 1945)	2	23	1	0	0	0	24
	*L. azevedoi* Navás, 1913	1	1	0	0	0	0	1
	*L. camposi* (Navás, 1933)	2	0	1	2	0	0	3
	*L. caucella* Banks, 1910	1	0	0	2	0	0	2
	*L. lateralis* Navás, 1913	1	17	0	0	0	0	17
	*L. maculosa* de Freitas & Penny, 2001	3	1	13	7	0	0	21
	*L. nigrovaria* (Walker, 1853)	2	1	10	0	0	0	11
	*L. squamisetosa* de Freitas & Penny, 2001	1	1	0	0	0	0	1
	*L. tarini* (Navás, 1924)	2	13	1	0	0	0	14
	*Leucochrysa* sp. 1	3	4	1	1	0	0	6
	*Leucochrysa* sp. 2	1	0	0	0	1	0	1
	*Leucochrysa* sp. 3	1	1	0	0	0	0	1
	*Leucochrysa* sp. 4	1	0	1	0	0	0	1
	*Leucochrysa* sp. 5	1	0	1	0	0	0	1
	*Leucochrysa* sp. 6	1	0	0	2	0	0	2
	Genus *Meleoma*							
	*M. macleodi* Tauber, 1969	2	0	0	1	3	0	4
	*M. titschacki* Navás, 1928	3	0	2	76	17	0	95
	*Meleoma* sp. 1	2	0	0	0	2	1	3
	Genus *Plesiochrysa*							
	*P. brasiliensis* (Schneider, 1851)	4	1	7	4	4	0	16
	*Plesiochrysa* sp. 1	1	0	0	0	5	0	5
	*Plesiochrysa* sp. 2	2	1	3	0	0	0	4
	Genus *Titanochrysa*							
	*T. annotaria* (Banks, 1945)	2	0	2	10	0	0	12
	*T. simpliciala* Tauber et al., 2012	1	0	0	2	0	0	2
	Genus *Ungla*							
	*Ungla* sp. 1	1	0	0	0	0	1	1
	*Ungla* sp. 2	1	0	0	0	1	0	1
Coniopterygidae	Genus *Coniopteryx* Species group *Scotoconioptery*							
	*C. fumata* Enderlein, 1907	1	0	1	0	0	0	1
	*C. josephus* Sarmiento & Contreras, 2019	1	1	0	0	0	0	1
	*C. latipalpis* Meinander, 1972	2	0	0	0	2	18	20
	*C. quadricornis* Meinander, 1982	2	3	8	0	0	0	11
	Species group *Coniopteryx*							
	*C. simplicior* Meinander, 1972	4	0	6	18	124	5	153
	*C. westwoodii* (Fitch, 1855)	2	0	3	0	2	0	5
	Genus *Conwentzia*							
	*C. barretti* (Banks, 1899)	3	0	0	1	2	17	20
	Genus *Neoconis*							
	*N. dentata* Meinander, 1972	5	3	3	18	10	1	35
	Genus *Semidalis*							
	*S. boliviensis* (Enderlein, 1905)	1	9	0	0	0	0	9
	*S. hidalgoana* Meinander, 1975	3	10	19	2	0	0	31
	*S. manausensis* Meinander, 1980	1	0	0	0	18	0	18
	*S. problematica* Monserrat, 1984	4	2	107	174	21	0	304
	*S. soleri* Monserrat, 1984	1	101	0	0	0	0	101
Hemerobiidae	Genus *Biramus*							
	*B. aggregatus* Oswald, 2004	1	0	0	18	0	0	18
	Genus *Hemerobiella*							
	*H. sinuata* Kimmins, 1940	1	0	0	1	0	0	1
	Genus *Hemerobius*							
	*H. alpestris* Banks, 1908	1	0	0	0	0	15	15
	*H. bolivari* Banks, 1910	4	0	4	14	8	2	28
	*H. discretus* Navás, 1917	3	0	0	1	31	286	318
	*H. domingensis* Banks, 1941	3	0	1	7	7	0	15
	*H. gaitoi* Monserrat, 1996	3	0	10	15	9	0	34
	*H. hernandezi* Monserrat, 1996	4	3	20	135	15	0	173
	*H. hirsuticornis* Monserrat & Deretsky, 1999	2	5	1	0	0	0	6
	*H. jucundus* Navás, 1928	5	2	3	16	69	268	358
	*H. martinezae* Monserrat, 1996	3	0	0	12	14	3	29
	*H. nigridorsus* Monserrat, 1996	1	0	0	2	0	0	2
	*H. withycombei* (Kimmins, 1928)	1	4	0	0	0	0	4
	Genus *Megalomus*							
	*M. minor* Banks, 1905	2	21	9	0	0	0	30
	*M. pictus* Hagen, 1861	2	0	0	0	2	1	3
	*Megalomus* sp. 1	1	0	0	0	0	2	2
	Genus Micromus							
	*M. subanticus* (Walker, 1853)	1	0	0	1	0	0	1
	Genus *Notiobiella*							
	*N. cixiiformis* Gerstaecker, 1888	1	0	0	0	1	0	1
	*N. mexicana* Banks, 1913	1	2	0	0	0	0	2
	Genus *Nusalala*							
	*N. championi* Kimmins, 1936	4	1	6	10	2	0	19
	*N. irrebita* (Navás, 1929d)	3	0	1	1	5	0	7
	*N. tessellata* (Gerstaecker, 1888)	1	1	0	0	0	0	1
	*N. unguicaudata* Monserrat, 2000	1	5	0	0	0	0	5
	Genus *Sympherobius*							
	*S. axillaris* Navás, 1928	2	0	0	0	2	8	10
	*S. distinctus* Carpenter, 1940	1	0	0	0	1	0	1
	*S. marginatus* (Kimmins, 1928)	3	0	0	4	1	1	6
	*S. similis* Carpenter, 1940	2	0	1	0	1	0	2
	*S. subcostalis* Monserrat, 1990	1	2	0	0	0	0	2
	*Sympherobius* sp. 1	1	0	0	0	0	1	1
Mantispidae	Genus *Dicromantispa*							
	*D. sayi* (Banks, 1897)	1	6	0	0	0	0	6
	Genus *Leptomantispa*							
	*L. pulchella* (Banks, 1912)	1	1	0	0	0	0	1
	Genus *Nolima*							
	*N. infensa* Navás, 1924	2	0	1	2	0	0	3
	*N. victor* Navás, 1914	1	0	0	3	0	0	3
	Genus *Zeugomantispa*							
	*Z. compellens* (Walker, 1860)	2	4	0	1	0	0	5
	*Z. minuta* (Fabricius, 1775)	3	0	2	2	4	0	8
Myrmeleontidae	Genus *Myrmeleon*							
	*M. immaculatus* De Geer, 1773	3	0	9	16	2	0	27
	*M. timidus* Gerstaecker, 1888	1	14	0	0	0	0	14
	*M. uniformis* Navás, 1920	2	0	0	2	4	0	6
	Genus *Ululodes*							
	*U. bicolor* (Banks, 1895)	1	1	0	0	0	0	1
	Genus *Ameropterus*							
	*A. trivialis* (Gerstaecker, 1888) *	1	1	0	0	0	0	1
Rhachiberothidae	Genus *Trichoscelia*							
	*T. santareni* (Navás, 1914)	3	2	5	1	0	0	8
	Total		329	393	737	438	630	2527
	Number of observed species	-	48	51	51	42	16	105
	Number of genera	-	20	18	20	18	8	28
	Sample completeness (%)	-	66%	74%	75%	88%	62%	73%

**Table 2 insects-13-00652-t002:** Total taxonomic and phylogenetic beta diversity (total dissimilarity [βsor]) as the sum of its components (turnover [βsim] and nestedness [βnes]) for the Neuropteran community along an elevational gradient of the Tacaná Volcano.

	Taxonomic Beta Diversity			Phylogenetic Beta Diversity		
Pair Sites	βsim	+βnes=	βsor	βsim	+βnes	=βsor
**1 vs. 2**	0.468	0.0217	0.489	0.333	0.007	0.34
**1 vs. 3**	0.659	0.0138	0.673	0.476	0.009	0.485
**1 vs. 4**	0.714	0.016	0.73	0.434	0.028	0.462
**1 vs. 5**	0.875	0.0615	0.936	0.5	0.211	0.711
**2 vs. 3**	0.372	0	0.372	0.255	0.004	0.259
**2 vs. 4**	0.476	0.0506	0.526	0.315	0.042	0.357
**2 vs. 5**	0.75	0.13	0.88	0.411	0.259	0.67
**3 vs. 4**	0.38	0.0599	0.44	0.289	0.047	0.336
**3 vs. 5**	0.5	0.261	0.761	0.323	0.296	0.619
**4 vs. 5**	0.25	0.336	0.582	0.117	0.336	0.453

## Data Availability

Data is contained within the article or supplementary material The data presented in this study are available in [39].

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
