# Peer review of "Diversity of Lacewings (Neuroptera) in an Altitudinal Gradient of the Tacaná Volcano, Southern Mexico"

_insects, 2022, doi:10.3390/insects13070652_

Round 1

Reviewer 1 Report

I read the manuscript by Rodolfo Cancino-López and co-authors with a great interest. The paper aims to present the patterns of alpha and beta diversity of Neuroptera in an altitudinal gradient of the Tacaná Volcano, Mexico.

The manuscript is well written, clear, well structured and relevant for the topics covered by the Special Issue. Reference list is relevant and recent publications are cited.

To my knowledge, there are almost no papers published on the altitudinal gradient of Neuroptera. The aims, methods, results and conclusion are thoroughly presented.

I have not found weakness of the paper.

The manuscript is exceptional and represents an outstanding contribution to the knowledge of the topics. I have no major comments, and I highly recommend the paper for acceptance.

Specific comments:

Page 1 Line 5: “Posgrado…” – Do you mean “Postgraduate study…”?

Page 11 Figure 4:

Two colours in columns (Black; Royal Blue or Navy Blue) have high saturation / intensity. It is almost impossible to distinguish them, therefore I suggest to use another colour combination.

Author Response

Page 1 Line 5: “Posgrado…” – Do you mean “Postgraduate study…”?

Answer: Yes, indeed “Posgrado en Ciencias Biológicas” is the official name of the graduate program in which Rodolfo Cancino has been enrolled.

Two colors in columns (Black; Royal Blue or Navy Blue) have high saturation / intensity. It is almost impossible to distinguish them, therefore I suggest to use another color combination. 

Answer: We intensified the colors to make the differentiation in the bars more evident. Thanks very much for your observations.

Reviewer 2 Report

Dear authors,

I detected many concerns on introduction, discussion and conclusions. You begin introduction writing about biodiversity patterns and ecological processes, the on elevation gradients, beta diversity, phylogenetic diversity and finally and poorly about Neuropterans. It would be fine if your manuscript was about biodiversity patterns, but, al your conclusions are about Neuropterans. So, introduction do not support conclusions. Regading discussion, I did not see a clear flow of the thematic. In fact, introduction and discussion are not concordant. You write about Neuropterans as your ms were about agronomy, but it is only about biodiversity, so I recommend you focus on it. Please try to fix all of these concerns. In sections of materials and methods and results, you could see in the attachment some suggestions.

Sincerely,

Author Response

Reviewer 2:

I detected many concerns on introduction, discussion and conclusions. You begin introduction writing about biodiversity patterns and ecological processes, the on elevation gradients, beta diversity, phylogenetic diversity and finally and poorly about Neuropterans. It would be fine if your manuscript was about biodiversity patterns but, all your conclusions are about Neuropterans.

Answer:

Thanks, we appreciate these observations.  We have added a broader context of previous Neuroptera studies in the introduction to provide proper and sufficient focus on the biological group and to make it concordant with the conclusions.

Regarding discussion, I did not see a clear flow of the thematic. In fact, introduction and discussion are not concordant. You write about Neuropterans as your ms were about agronomy, but it is only about biodiversity, so I recommend you focus on it.

Answer: Suggestions were taken into account so the manuscript is focused on the biodiversity of Neuroptera, agroecological aspects are mentioned at a minimum only in relationa to some agroecosystems working as biodiversity refuge in the study area (i.e., coffee plantations). We hope that with these changes the manuscript will improve consistency and clarity.

Observations made on the pdf file.

Answer: Thanks for taking the time to do a thorough revision of the manuscript text, we appreciate very much all your observations, so the vast majority of the comments and suggestions written over the pdf file were taken into account, however:

  • About the section where it is requested to indicate the order and family of lichens and grasshoppers (in the introduction), the text was left as it was because a large number of families and orders were listed in that study, so in order to be more synthetic we did not change this.
  • About moving table 1 to the appendices section, we consider that it is more informative and illustrative to have this table in the main text; taxonomy and survey work were also crucial in this work, so we prefer to be as comprehensive and specific as possible regarding species of Neuroptera present and their numbers.
  • About table 2 being very confusing, we preferred not to make similarity matrices, as we would need to make six matrices (one for each component), however we made accordingly some modifications to this table in order to make it more informative and clear.
  • Regarding inventory completeness, it was suggested to remove this section altogether, as it is not relevant to the study goals. Nonetheless, we believe it is important to include this section to highlight the importance of good inventories and sampling methods in this type of study, particularly for Neuroptera, a group that requires an intense sampling effort.
  • About old (75 years) references, indeed we acknowledge there were many references that did not fit with goals of the manuscript, so as suggested, the number of old citations or those that are not associated with the manuscript goals were reduced, with one exception, Pérez-Toledo et al. 2021, because although only “Mexico” is mentioned in the title, the article was carried out on a mountain in southeastern Mexico, similar to our study.

Reviewer 3 Report

A very interesting paper on neuropteran diversity at different altitudes on the Tacaná Volcano in Mexico, the research has been thoroughly undertaken and the interpretations and conclusions are well backed up by their data and analysis. Below are some minor comments for the authors, and some suggestions to improve the English of the manuscript.

I was confused to see that a paper by the first author on a very similar topic, using the same data(?) has not been referenced in this manuscript?

 Paper: Cancino-Lopez, Martins, Contreras-Ramos, 2021. Neuroptera Diversity from Tacaná Volcano,Mexico: Species Composition, Altitudinal and Biogeographic Pattern of the Fauna. Diversity, 2021, 13, 537.

 This paper needs referencing and also it needs mentioning that this paper uses the same data and sites.

Comments/suggestions below:

Simple Summary:

 Line 14: “…. little information on their patterns and distribution” Is this world-wide or just in Mexico?

 Material and Methods:

 Sampling  method:

Other than altitude, was there any other specific reasons that the sites were chosen? Also what was your reasoning for selecting the specific altitudes for the study?

For the sweeping - was this done at the same time everyday? and was it in one 4 hour block or at different times throughout the day that totaled 4 hours?

Did you focus on particular habitats or did you not limit yourself to a particular habitat?

 Line 141: a.s.l. please define what this means in the text the first time it is used, e.g. above sea level (a.s.l)

 With regards to Table S1, is there a reason that this is put into the supplementary data and not the main text?

 Figure 1:

 The two maps should be labelled A (Geographical Location) and B (Sites of study). Also, could you maybe pattern the habitat types (meadow and forest); so that they can be separated on something other than color?

 Figure 2:

 Would it be possible to replace A-F with the site names (i.e. Site 1, Site 2 etc..)?

 Formatting/appearance: The font appears different to the text in the rest of the manuscript, also the resolution appears low quality.

 Table 2:

 Formatting: headings βnes and βsor are not in bold on the Beta phylogenetic diversity side

 Figure 3:

 In the figure and in the main-text: the figure has abbreviations TDiv, TDis, and TVar, however in the main text (Data analysis section, and Alpha diversity: Species and taxa section) these are written as DivT, DisT and VarT – please change the figure or the abbreviations in the main text to be consistent.

 In the main-text it would be good to point the reader to the graph you are talking about, e.g. using Figure 3A, instead of just referring to Figure 3.

 Formatting: instead of using 3 colors on the graphs could you use 3 symbols, it makes it easier for the reader to differentiate.

 Figure 4:

 Is it possible to use two more different colors, or patterned fill to easily differentiate between βsim and βnes? Also, could you put a title on each of the graph to enable a quick view of what they represent? There also appears to be parts of a box around each graph, which could do with deleting.

 Discussion:

 Do not forget to reference your figures in this section, to help illustrate.

 Lines: 324-325, 327:

 You list q0, q1, q2 diversity – have q0, and q1 diversity been defined in the text previously?

A paper not referenced that may be of interest:

 González-Reyes, Corronca and Rodriguez-Artigas, 2017. Changes of arthropod diversity across an altitudinal ecoregional zonation in Northwestern Argentina. PeerJ; 5; e4117.

 Suggested English changes:

 Simple Summary:

Lines: 12 -14:

Lacewings are insects with a great diversity of shapes in morphology and habits, and with are potentially importantce as bioindicators and biological control agents,. However, despite this there is little information on their patterns of distribution and diversity.

Lines: 15 – 18:

We analyzed Neuroptera diversity locally and between sites through an elevation gradient, finding two marked trends: 1) a decrease in richness with increasing altitude and, 2) greater diversity and potential richness at an intermediate elevation.

Abstract:

Line: 23:

Neuroptera is a group an order of insects

Lines: 33-35:

 The highest total beta diversity value was recorded at elevations above 3000 m, while highest number of species and supra-specific taxa were observed at sites between 600 and 2000 m, with an evident decrease above 3000 m.

 Results:

Lines: 211 -214:

The highest number of species was presented represented by Chrysopidae (51 species) and Hemerobiidae (29 species), and the greatest abundance by Hemerobiidae (709 individuals) and Coniopterygidae (1094 individuals) (Table 1).

Table 1:

Rachiberothidae should be Rhachiberothidae

Alpha diversity: Species and taxa

Lines 243-245:

The taxonomic distinctness (DivT) in the first four sites presented similar values,. The with the highest value is observed in site three, and the lowest value, in site five.

Line 246: You have two spaces before “Finally, ….”

Discussion:

Line 292: Rachiberothidae should be Rhachiberothidae

Lines 296-298:

This agrees with what was observed in a study on elevational diversity patterns with of Chrysopidae, recording suggesting that temperature had a significant effect on the abundance of these green lacewings [75], reducing its their abundance in areas with lower temperatures.

Lines 306-307:

Based on previous studies, the specificity to a plant substrate could explain the presence of dominant species at specific sites, so that sites at different levels that shared dominant species also had the same type of vegetation.

Discussion:

Line 308 (and elsewhere in the text, line 412, 423)

“In the volcano”, replace with “On the volcano”

Line 336-337:

This allowed means that at high altitudes, the richness and diversification of the Neuroptera are exclusive to those species adapted to extreme conditions.

Lines 343-347:

The values of the alpha taxonomic distinctness showsed that the sites between 600 and 1700 m had the highest values,. This which showed suggests that these areas had greater diversity in their taxonomic structure, as a reflection of greater phylogenetic separation between the species that make up these communities. And with The high values of taxonomic variation, reflecting that most species are concentrated in a few supra-specific taxa [61,86,87].

Line 363:

As an example of this, the family Chrysopidae family with has greater diversity at low and medium……..

Lines 381-385:

When we compared diversity between sites, almost all comparisons were explained by species turnover, except for the pair  two sites above 2000 m, which can be explained mainly by nestedness. Previously, it was this has been recorded in Chrysopidae in an altitudinal study, where nestedness replaced turnover as the main component of dissimilarity as elevation increased (mainly in sites with low temperatures) [75].

Line 392:

… more important [92], which leads us to think if suggests this tendency could be repeated for certain

Line 411:

… values were low compared to species-level values.,  Ddue to the low supra-generic diversity ….

Line 417:

Appears to be 2 spaces before “This means ..”

Line 424:

“And with the medium and high altitudes with the highest abundance peaks” needs rewording e.g. “ The highest abundance peaks were observed at medium and high altitudes

Line 428-431:

The highest estimated species richness was recorded at low altitudes, decreasing with increasing elevation. The values of diversity q1 and q2 with have similar trends, observing with a decrease with increasing altitude, and the highest value at the site above 1000 m; the lower diversity value at low altitudes (>1000 m) is possibly due

Line 440:

recorded a similar trend to that calculated for species., Bbut with much lower turnover

Line 444:

limitations (reflected by high turnover rates), as it is mentioned in the study by Fontana

Author Response

We made most of the changes and suggestions that were indicated.

  • This paper needs referencing and also it needs mentioning that this paper uses the same data and sites. Answer: The manuscript by Cancino et al. 2021 was cited, however it is specified that only some data were incorporated into the analyses of this paper.
  • Other than altitude, was there any other specific reasons that the sites were chosen? Also what was your reasoning for selecting the specific altitudes for the study? For the sweeping - was this done at the same time everyday? and was it in one four hour block or at different times throughout the day that totaled 4 hours? Did you focus on particular habitats or did you not limit yourself to a particular habitat? Answer: For the selection of the sampling sites, altitude was primarily taken into account, but also aspects such as whether they were sites with native vegetation and not disturbed, as well as different vegetation types along the gradient. Also, we considered sites where the local people could allow us access and field work. In the case of sweeping, the conditions and times for this technique of sampling are now explained in the manuscript.
  • With regards to Table S1, is there a reason that this is put into the supplementary data and not the main text? Answer: There is no particular reason other than to avoid the saturation of tables and information in the main text, besides this information may still be easily seen in the supplementary material.
  • A paper not referenced that may be of interest: González-Reyes, Corronca and Rodriguez-Artigas, 2017. Changes of arthropod diversity across an altitudinal ecoregional zonation in Northwestern Argentina. PeerJ; 5; e4117. Answer: This is a very interesting reference, which I will take into account in future work, currently it is set aside because the geographical expansion (spatial scale) is greater than the one that was worked on in this study and is not comparable with the study area we worked on.
  • Suggested English changes. Answer: We greatly appreciate the suggestions provided to improve the English of the manuscript.

Round 2

Reviewer 2 Report

I read a revised ms, however some amendments are needed. Introduction does not match with your current and well written discussion. Your introduction is focused in biodiversity and your discussion is on Neuroptera. I think that a ms submitted to that special  issue  on Neuroptera must be abundant in biology, ecology and the indicator value of that group. However, you write much about biodiversity, phylogenetic diversity, beta diversity and elevational gradients. Please make a correct link between both sections. Figures (species rank curves) and the inventory table need to be corrected as I previously indicated you.

Author Response

Reviewer 2:  I read a revised ms, however some amendments are needed. Introduction does not match with your current and well written discussion. Your introduction is focused in biodiversity and your discussion is on Neuroptera. I think that a ms submitted to that special  issue  on Neuroptera must be abundant in biology, ecology and the indicator value of that group. However, you write much about biodiversity, phylogenetic diversity, beta diversity and elevational gradients. Please make a correct link between both sections. Figures (species rank curves) and the inventory table need to be corrected as I previously indicated you.

Dear reviewer, thank you very much for your valuable comments and observations.

Regarding the approach of the introduction to the study group (Neuroptera), changes were made in order to provide emphasis on the group, not leaving aside general information about the patterns of diversity, which are an essential component of this research (i.e., diversity patterns of Neuroptera in an altitudinal gradient). We hope that with these modifications, the introduction is better aligned with the manuscript’s objectives, discussion, and conclusions.

The figures and table are now being submitted in high quality. We followed the suggested changes: we have consolidated the mosaic of figures into a single graph to facilitate the comparisons and made the corresponding modifications in the table, following the specific suggestions made to the previous version of the manuscript.

Round 3

Reviewer 2 Report

I read an improved ms.

Congrats